# Prognostic and Immunological Role of *STK38* across Cancers: Friend or Foe?

**DOI:** 10.3390/ijms231911590

**Published:** 2022-09-30

**Authors:** Yankuo Liu, Zhiyuan Shi, Zeyuan Zheng, Jinxin Li, Kunao Yang, Chunlan Xu, Qing Liu, Zhicheng Gong, Yi Yang, Yue Zhao, Zuodong Xuan, Huimin Sun, Chen Shao

**Affiliations:** 1Department of Urology, Xiang’an Hospital of Xiamen University, School of Medicine, Xiamen University, Xiamen 361101, China; 2Central Laboratory, Xiang’an Hospital of Xiamen University, School of Medicine, Xiamen University, Xiamen 361101, China

**Keywords:** *STK38*, pan-cancer, immune infiltration, prognosis, survival

## Abstract

Although *STK38* (serine-threonine kinase 38) has been proven to play an important role in cancer initiation and progression based on a series of cell and animal experiments, no systemic assessment of *STK38* across human cancers is available. We firstly performed a pan-cancer analysis of *STK38* in this study. The expression level of *STK38* was significantly different between tumor and normal tissues in 15 types of cancers. Meanwhile, a prognosis analysis showed that a distinct relationship existed between *STK38* expression and the clinical prognosis of cancer patients. Furthermore, the expression of *STK38* was related to the infiltration of immune cells, such as NK cells, memory CD4+ T cells, mast cells and cancer-associated fibroblasts in a few cancers. There were three immune-associated signaling pathways involved in KEGG analysis of *STK38*. In general, *STK38* shows a significant prognostic value in different cancers and is closely associated with cancer immunity.

## 1. Introduction

Cancer is a leading cause of death and contributes to the poor quality of life of patients in many countries worldwide, and the incidence and mortality of it are significantly elevated globally [1]. Although we have made great efforts in the early diagnosis and the treatment of multiple cancers to decrease the incidence rate, the mortality of malignant tumors still has a high value [2]. Phenotypic and genetic alterations in cancer cells contribute to tumor heterogeneity and therapy resistance. Moreover, the same gene may play disparate roles across different cancers [3]. Therefore, an integrative pan-cancer analysis of cancer-related genes is essential for understanding their roles in cancer progression. The tumor microenvironment (TME) is an essential part of cancer. Novel targets of the TME have been demonstrated that can strengthen the effect of a variety of cancer therapies, particularly immunotherapies which work by promoting the host’s antitumor immune responses [4]. The TME is a sophisticated and multilevel network of interactions between cancer cells and the surrounding components, including immune cells, endothelial cells, fibroblasts and stromal cells. Immune cells play a pivotal role when they are compared with other cellular components. The dynamic tumor-immune cell interactions can secrete cytokines and growth factors, thereby contributing to tumor cell proliferation, metastasis and the inhibition of an antitumor immunity response [5,6]. Hence, exploring the relationship between gene expression and immune cell infiltration can help us to find potential immunotherapy targets.

As members of a subgroup of the AGC (protein kinase A/G/C, PKA/PKG/PKC-like) group of protein kinases, the NDR (nuclear Dbf2-related)/LATS (large tumor suppressor) family of kinases have drawn much attention in the past few years. As four members of the NDR/LATS protein kinase family, NDR1 (*STK38*), NDR2 (*STK38*L), LATS1 and LATS2 are encoded by the mammalian genome. Like most AGC protein kinases, the NDR/LATS family sequences are highly conserved from yeast to humans. In yeast and invertebrates, the NDR/LATS family play an independent role in the regulation of an extensive cellular processes, including apoptosis and cell proliferation [7,8]. Emerging studies have demonstrated that yeast NDR/LATS kinases are involved in controlling the mitotic exit network and septation initiation network in budding and fission yeast, respectively. In *Drosophila melanogaster*, the fly NDR/LATS kinases are involved in the regulation of neurite outgrowth and morphology [9]. In mammals, four NDR/LATS kinases can manifest overlapping or completely different functions.

*STK38* (serine-threonine kinase 38) was originally found in the nuclei, while past studies demonstrated that it was also a cytoplasmic kinase [10]. Three significant structures were found in the STK38 protein, including a central kinase catalytic domain, a conserved N-terminal regulatory domain (NTR) and a C-terminal hydrophobic motif. As a protein kinase, *STK38* has been proved to regulate various biological processes, including DNA damage responses, centrosome duplication, cell cycle progression and apoptosis [11]. Furthermore, previous studies have also demonstrated that *STK38* participants in neurodevelopment [12], embryonic development [13], and cancer biology [14]. *STK38* plays inconsistent roles in different cancers. Some studies have demonstrated the oncogenic potential of the *STK38* gene in progressive ductal carcinoma in situ in the breast tissue [15], lung adenocarcinoma [16,17] and ovarian cancer [18]. On the contrary, *STK38* could reduce the proliferation and colony formation of glioblastomas [19]. *STK38* was also discovered to inhibit the metastasis of prostate cancer both in vitro and in vivo [20].

## 2. Results

### 2.1. STK38 Expression Analysis in Human Normal Tissues and Cancers

In our study, we intended to perform a comprehensive analysis of *STK38* in various cancers. In our initial analysis, the phylogenetic tree was made to explore evolutionary relationship of the STK38 protein across the studied species (Appendix A). To explore the expression difference of *STK38* between cancer tissues and normal tissues, we used the TIMER2.0 web tool. As displayed in Figure 1A, the *STK38* expression was upregulated in cholangiocarcinoma (CHOL), esophageal carcinoma (ESCA), glioblastoma multiforme (GBM), kidney renal papillary cell carcinoma (KIRP), liver hepatocellular carcinoma (LIHC), lung adenocarcinoma (LUAD), lung squamous cell carcinoma (LUSC), stomach adenocarcinoma (STAD) and uterine corpus endometrial carcinoma (UCEC). On the contrary, a low expression of *STK38* was discovered in four cancers: breast invasive carcinoma (BRCA), kidney chromophobe (KICH), colon adenocarcinoma (COAD) and prostate adenocarcinoma (PRAD). Notably, the *STK38* expression in skin cutaneous melanoma (SKCM) primary tissue was lower than it was in the respective metastatic tumor tissue.

The “Pathological Stage Plot” module from GEPIA2 was used to analyze the relationship between *STK38* expression and the pathological staging in different tumors, which demonstrated that there were significant stage-specific expressional changes in some cancers including adrenocortical carcinoma (ACC), LIHC, ovarian serous cystadenocarcinoma (OV), pancreatic adenocarcinoma (PAAD), thyroid carcinoma (THCA), UCEC, SKCM (Figure 1B, all *p* < 0.05), while in other cancers, we found no clear relationship.

In addition to the transcription, we also used the UALCAN portal to assess the STK38 protein expression in different cancers based on the CPTAC dataset. The expression of the STK38 total proteins was upregulated in PAAD, ccRCC, and LIHC when they were compared to that of the normal samples. A lower STK38 expression in the tumors was discovered in ovarian cancer, COAD and LUAD. Moreover, we further evaluated the IHC results from the HPA and combined the results with the STK38 protein expression profile from the CPTAC. As displayed in Figure 2, the results from the HPA analysis were consistent with those from the CPTAC analysis.

### 2.2. Survival Prognosis Analysis of STK38 in Cancers

Cancer patients were divided into high- and low-expression groups, and then we analyzed the correlation between the *STK38* expression levels and the clinical outcome of patients with different cancers. As shown in Figure 3, a low expression of *STK38* was related to a poorer overall survival (OS), including in kidney renal clear cell carcinoma (KIRC), rectum adenocarcinoma (READ) and thymoma (THYM). Moreover, a low *STK38* expression was also found to be associated with a poor disease-free survival (DFS) in KIRC. Moreover, highly expressed *STK38* was also correlated with a poorer OS for brain lower grade glioma (LGG) and DFS for ACC, LGG and LIHC (all log rank *p* values < 0.05).

Furthermore, we utilized the Kaplan–Meier plotter to conduct a survival prognosis analysis in the breast, gastric, liver, lung and ovarian cancer tissues. As shown in Appendix A, a high expression of *STK38* was correlated with the poor OS, PPS, FP and PFS in gastric cancer. Moreover, a high level of *STK38* expression was related to a poor OS, RFS and PFS for liver cancer. On the contrary, a low *STK38* expression was associated with a poor OS and PFS prognosis for ovarian cancer. Nevertheless, there were a complicated situations in breast cancer and lung cancer. A highly expressed level of *STK38* was related to a poor PPS and DMFS for breast cancer and poor OS and FP for lung cancer, however, a lowly expressed *STK38* was linked to a poor OS and RFS for breast cancer and a poor PPS for lung cancer. The above data indicated that the expression of *STK38* is differentially associated with the clinical outcome of the patients that are bearing different cancers.

### 2.3. Genetic Alteration and DNA Methylation Analysis of STK38

DNA methylation and genetic mutations are widely involved in tumor initiation, development and metastasis. We firstly analyzed the genetic mutation status of *STK38* in the cancer patients in the TCGA database. The highest mutation frequency of *STK38* happened in the uterine cancer patients, and the “mutation” accounted for the majority of all the alterations. The “amplification” was the majority alteration type in the OV, CHOL, esophageal adenocarcinoma, lymphoid neoplasm diffuse large B-cell lymphoma (DLBC), sarcoma (SARC), uterine carcinosarcoma (UCS), KIRP and GBM tissues (Figure 4A). The distribution of the *STK38* genetic alteration is presented in Figure 4B. We found that majority of the mutation sites are located in the central kinase catalytic domain. Figure 4C shows the protein structure of STK38. Moreover, we analyzed the possible relationship between the *STK38* alteration and the clinical outcome. The result demonstrated that the UCEC patients carrying a mutated version of *STK38* revealed that there was no relationship with this and their overall survival (*p* = 0.212), disease-free survival (*p* = 0.242), progression-free survival (*p* = 0.163) and disease-specific survival (*p* = 0.123), in comparison with patients without the *STK38* mutation (Figure 4D). As one of significant epigenetic modifications, DNA methylation can promote the silencing of tumor suppressor genes and thus, this results in tumorigenesis and the progression of the cancer [21,22]. We found that STK38 expression was significantly correlated with the DNA methylation level in 24 kinds of tumors (Table 1). *STK38* expression was negatively associated with promoter methylation levels in 15 cancers. We show the six strongest negative correlations (cervical squamous cell carcinoma and endocervical adenocarcinoma (CESC), COAD, ESCA, READ, THCA and UCS) in Appendix A. *STK38* expression was positively associated with the promoter methylation levels in eight cancers. We present the three strongest positive correlations (pheochromocytoma and paraganglioma (PCPG), THYM and testicular germ cell tumors (TGCT)) in Appendix A.

### 2.4. STK38 Is Correlated with the TMB and MSI

In BRCA, LGG, ACC, LUAD and SARC, *STK38* expression was positively associated with the TMB, whereas in COAD, THYM and UCEC, it was adversely connected with the TMB (Appendix A). *STK38* expression was inversely correlated with the MSI in COAD, DLBC and KIRP, but it was positively connected to it in LUSC and LUAD (Appendix A).

### 2.5. STK38 Expression Correlates with Tumor Immune Infiltration

Tumor-infiltrating immune cells which were significant components of the TME have been demonstrated to be closely related to the initiation, progression and metastasis of malignant cancers [23,24]. To explore the association between immune cell infiltration and the *STK38* expression level in different cancers, the RNA-sequencing expression profiles and corresponding clinical information that is related to *STK38* were downloaded from the TCGA dataset. According to existing data, we utilized the “CIBERSORT” algorithm to evaluate the 22 immune cells. We found that the expression level of *STK38* was positively correlated with infiltrating levels of the NK cells and mast cells, however, it was negatively correlated with the CD4+ T cells. (Figure 5A).

Cancer-associated fibroblasts in the TME were proven to take part in the modulating of the function of various tumor-infiltrating immune cells [23,24]. We used a variety of algorithms to analyze the potential relationship between the infiltration level of cancer-associated fibroblasts and *STK38* gene expression in different cancer types of TCGA. We found that there is a positive correlation between the infiltration level of the tumor-associated fibroblasts and expression of *STK38* for head and neck squamous cell carcinoma-HPV-(HNSC-HPV-), LGG, PAAD, PRAD, SARC, TGCT and UCS as based on all or most of the algorithms (Figure 5B). Moreover, we observed a negative correlation between the infiltration level of the tumor-associated fibroblasts and expression of BRCA-Her2, ESCA, HNSC-HPV+ and STAD. The scatter plots of the abovementioned tumors which were generated using one algorithm are shown in Figure 5C. For instance, the *STK38* expression level of LGG is positively correlated with the status of the cancer-associated fibroblasts infiltration (Figure 5C, cor = 0.309, *p* = 4.93 × 10^−12^). 

Immune cells and stromal cells are two major types of non-tumor components in the TME, and they have been proposed to be a prognostic biomarker of cancer patients. The higher score that was calculated using ImmuneScore or StromalScore indicated the larger proportion of the immune or stromal components in the TME. The ESTIMATEScore was the sum of ImmuneScore and StromalScore, thereby representing the comprehensive proportion of both of the components in the TME. The top three tumors that were most obviously associated with the expression of STK38 were PRAD, PAAD and TGCT (StromalScore), LGG, UVM and PAAD (ImmuneScore), and LGG, PAAD and DLBC (ESTIMATEScore) respectively (Figure 6), which indicates that high STK38 expression is correlated with high stromal components or immune components in these cancers. The results demonstrated the strong relationship between STK38 expression and immune infiltration level in several cancers.

### 2.6. Enrichment Analysis of STK38-Related Partners

To deeply explore the molecular biological mechanism of *STK38* in cancer initiation and progression, we obtained the STK38-interacting proteins and the *STK38* expression correlated genes to perform pathway enrichment analyses. Using the STRING website, we observed 50 experimentally detected STK38-binding proteins. Figure 7A demonstrates the interaction network of these proteins. After that, we combined all of the tumor expression data of TCGA using the GEPIA2 tool and obtained the top 100 genes which were associated with *STK38*. The expression of *STK38* was found to be positively correlated with that of the *MAPK14* (Mitogen-Activated Protein Kinase 14) (R = 0.56), *SRPK1* (SRSF Protein Kinase 1) (R = 0.66), *SP1* (Sp1 Transcription Factor) (R = 0.5), *NUMB* (NUMB Endocytic Adaptor Protein) (R = 0.43) and *PKN2* (Protein Kinase N2) (R = 0.51) genes (all *p* < 0.001) (Figure 7B). The heatmap showed that *STK38* had a significantly positive relationship with five genes in most of the cancers (Figure 7C). Then we combined these two datasets to conduct the GO and KEGG enrichment analyses. As shown in Figure 7D, three pathways including the “Toll-like receptor signaling pathway”, “RIG-I-like receptor signaling pathway”, “TNF signaling pathway” were identified as the most importantly enriched KEGG pathways, thereby suggesting that *STK38* was closely related to immunity in the tumors.

## 3. Discussion

STK38 is a member of the NDR/LATS kinase family, and it is mainly located in the cell nuclei. In addition to it having a central kinase catalytic domain, STK38 has a N-terminal regulatory domain (NTR) and a C-terminal hydrophobic motif. STK38 kinase can play opposing roles in tumors by serving as a tumor suppressor or an oncogenic factor in different conditions. On the one hand, by positively regulating the cell proliferation and centrosome duplication, *STK38* displayed oncogenic characteristics at the cellular level. Previous studies have reported that high *STK38* expression was associated with centrosome overduplication [25], thus contributing to chromosomal instability. Moreover, *STK38* knockdown significantly decreased the MYC levels and promoted apoptosis, leading to the growth inhibition of MYC-addicted human B-cell lymphoma tumors [26]. On the other hand, *STK38* provided a tumor suppressive capacity by phosphorylating YAP, thereby promoting its cytoplasmic retention and subsequent degradation [27]. Despite emerging biological experiments that are providing the correlation between *STK38* and a few tumors, no pan-cancer analysis about *STK38* in different cancers is available to date.

We firstly analyzed the expression of *STK38* in various of cancers using the TCGA and CPTAC datasets. The results displayed that when they were compared to normal tissue samples, a low *STK38* expression level was found in BRCA, COAD, KICH and PRAD, whereas a high expression was observed in nine types of cancers, including UCEC, CHOL, GBM, KIRP, LIHC, LUSC, LUAD, STAD and ESCA. The result from the CPTAC analysis showed that a higher STK38 protein expression in cancer was discovered in PAAD, ccRCC and LIHC, and the expression of STK38 total proteins was lower in ovarian cancer, COAD and LUAD when they were compared to the normal tissues. Additionally, the CPTAC result was consistent with that of the IHC analysis data. A high *STK38* expression was correlated with better OS and DFS prognosis in KIRC, which demonstrates that *STK38* may serve as a protective factor in this cancer. However, a high STK38 expression was correlated with a poor clinical outcome, particularly in patients with LGG, gastric cancer and liver cancer, based on the Kaplan–Meier analysis. Past studies have reported that the *STK38* expression was higher in primary prostate cancer than it was in metastasis cancer, and that it could be a potential tissue biomarker during tumor progression [20]. These results indicate that *STK38* is a promising biomarker to estimate the clinical outcome of cancer patients.

Furthermore, we found that *STK38* plays a significant role in cancer immunity. The cancer immune status is discovered to be closely related to the infiltrating concentration and composition of the immune-related cells in the TME [28,29,30]. Past studies have demonstrated that *STK38* is correlated with innate and adaptive immune responses. As it is downstream of MST1, *STK38* was found to regulate the egress of mature thymocytes from the thymus and the interstitial migration of naive T cells [31]. *STK38* increases the virus-induced production level of type I interferon and proinflammatory cytokines in a kinase-independent manner [32]. However, the correlation between *STK38* expression and cancer immunity remains unclear.

The ESTIMATE algorithm was shown to be effective and convenient to evaluate the tumor purity and serve as a clinical prognostic factor in human cancers [33]. We screened out the top three cancers with the strongest correlation between the *STK38* expression and the ImmuneScore or StromalScore based on the TCGA cohort. The result showed that *STK38* was significantly positively associated with the stromal component of the TME in PRAD, PAAD and TGCT and the immune component in LGG, UVM and PAAD. Furthermore, the KEGG pathway analysis demonstrated that *STK38* was strongly correlated with the immune-related pathways, including Toll-like receptor signaling pathways (TLRs). TLRs are a significant fraction of the pattern recognition receptors (PRRs) which play an essential role in the innate immune response. TLRs were reported to active the downstream pathways and are related to the initiation of immune diseases and cancers [34,35]. Meanwhile, previous study has demonstrated that *STK38* may downregulate TLR9 signaling to decrease the inflammatory cytokine production and protect the host from an inflammatory injury during infection [36]. Furthermore, the result in our study indicates that *STK38* is negatively associated with infiltrating levels of CD4+ T cells, and it is positively correlated with NK cells and Mast cells in most cancer types. These findings suggest that *STK38* plays a complicated role in regulating immunity.

Cancer-associated fibroblasts (CAF) express more biomarkers than normal fibroblasts do, including FAP and αSMA, which have been used as biomarkers to isolate CAF populations from the tumor tissue [37]. A previous study has demonstrated that CAFs have a pro-tumorigenic function and participate in regulating tumor metastasis by secreting growth factors and remodeling the extracellular matrix [37]. In our study, the result showed that *STK38* expression was significantly positively associated with the infiltration level of CAFs in HNSC-HPV-, LGG, PAAD, PRAD, SARC, TGCT and UCS (appeared in four algorithms), suggesting that *STK38* may promote the transformation and activation of CAFs in these cancers. The TMB is closely associated with the efficacy of immunotherapy in cancer. Previous studies have displayed that a high TMB is related to a better prognosis of patients that are treated with immune checkpoint inhibitors in melanoma [38], urothelial carcinoma [39] and head and neck cancer [40]. The MSI is also an important predictor of the ICI response. In our study, we found that *STK38* expression is associated with the TMB in eight kinds of cancers and with *STK38* expression being associated with the MSI in five kinds of cancer. Therefore, *STK38* may be a potential biological marker for the treatment effect of immunotherapy in the above-mentioned cancers.

However, there are several limitations in our study. First, although the pan-cancer analysis has provided us with some important insights on *STK38* in tumors, it is essential to conduct related experiments to verify our results. Next, we will perform biological experiments to explore the mechanism of *STK38* in different cancers. Second, the result showed that expression of *STK38* was associated with immune cell infiltration and the clinical prognosis in cancers, we cannot verify whether *STK38* may affect patient clinical endpoint by the immune pathways.

In summary, based on our comprehensive pan-cancer analysis of *STK38*, we found a statistical correlation between the *STK38* expression and patient survival, immune cell infiltration, cancer-associated fibroblasts, tumor mutation burden and microsatellite instability in several human cancers, thereby contributing to further our understanding of the roles of *STK38* in cancer.

## 4. Materials and Methods

### 4.1. Gene Expression Analysis

We used TIMER2 (tumor immune estimation resource, version 2, http://timer.cistrome.org/, accessed on 20 February 2022) to analyze the expression profile of *STK38* between tumor types and the adjacent normal tissues. Additionally, the *STK38* expression in various of pathological stages of all TCGA tumors was obtained by using the “Pathological Stage Plot” module in GEPIA2 (http://gepia2.cancer-pku.cn/, accessed on 21 March 2022). The log2 [TPM (Transcripts per million) +1] transformed expression data were applied for the box or violin plots. The UALCAN tool (http://ualcan.path.uab.edu/analysisprot.html, accessed on 8 April 2022) was used to explore the cancer omics data. We obtained protein expression data of STK38 by using the CPTAC dataset [41]. The STK38 total protein expression levels were found in tumor and normal tissues.

### 4.2. Immunohistochemistry (IHC) Staining

To assess the differences in STK38 expression at the protein level, we downloaded and analyzed IHC images of STK38 protein expression in six tumors tissues and normal tissues, including KIRC (kidney renal clear cell carcinoma), PAAD (pancreatic adenocarcinoma), LIHC (liver hepatocellular carcinoma), ovarian cancer, COAD (colon adenocarcinoma) and LUAD (lung adenocarcinoma) from the HPA (Human Protein Atlas) (http://www.proteinatlas.org/, accessed on 17 March 2022).

### 4.3. Survival Prognosis Analysis

The OS (Overall Survival) and DFS (Disease-Free Survival) significance map data and survival plots of *STK38* across all TCGA tumors were obtained using GEPIA2. We used cutoff-high (50%) and cutoff-low (50%) values to divide the high expression and low expression cohorts [42]. The log-rank test was used in the hypothesis testing.

We used the Kaplan–Meier plotter (http://kmplot.com/analysis/, accessed on 24 March 2022) to analyze the *STK38* expression that was associated OS, post-progression survival (PPS), first progression (FP), progression-free survival (PFS), relapse-free survival (RFS) and the distant metastasis-free survival (DMFS) outcomes of patients [41]. Cancer patients were divided into two groups through choosing the “auto select best cutoff” setting. Then we obtained Kaplan–Meier survival curves, and the log-rank *p*-values and the HRs were analyzed.

### 4.4. Genetic Alteration Analysis

We used the cBioportal tool (http://www.cbioportal.org/, accessed on 17 March 2022) to calculate the *STK38* alteration frequency and gene mutation profile in cancer patients [43]. Microsatellite instability (MSI) is associated with the nucleotide insertions and deletions in the microsatellite loci [44]. We collected the TMB (Tumor mutational burden) and MSI scores in different cancers from TCGA.

### 4.5. Immune Infiltration Analysis

We used the CIBERSORT algorithm to evaluate the association between immune infiltration and *STK38* expression level in different cancers. Cancer-associated fibroblasts were used for evaluating the immune infiltration level by using TIDE, TIMER, CIBERSORT, EPIC, QUANTISEQ, MCPCOUNTER and XCELL algorithms. The data are shown as a heatmap and scatter plots. ESTIMATE is an algorithm that is used to evaluate the degree of infiltration of the immune and stromal cells in tumors [45]. We used the ESTIMATE algorithm to calculate the immune, stromal and ESTIMATE scores for different tumors. Additionally, the association between *STK38* expression and these scores was evaluated using the R software package (version 2.2.10) “estimate”.

### 4.6. STK38-Related Gene Enrichment Analysis

The STRING website (https://string-db.org/, accessed on 11 January 2022) was used for the query of a single protein name (“*STK38*”) and organism (“Homo sapiens”). The main parameters were: the minimum required interaction score [“Low confidence (0.150)”], the meaning of the network edges (“evidence”), the max number of the interactors to show (“no more than 50 interactors” in 1st shell) and the active interaction sources (“experiments”). Then we obtained the available experimentally determined STK38-binding proteins. GEPIA2 was used to pick up the top 100 STK38-correlated targeting genes from the datasets of all TCGA cancer and normal tissues. We used the “correlation analysis” module of GEPIA2 for a pairwise Pearson correlation analysis of *STK38* and the related genes. We applied the log2 TPM to produce the dot plot. In addition, we used the “Gene_Corr” module of TIMER2 to image the heatmap data of the selected genes. In the end, we uploaded these gene lists onto DAVID website (https://david.ncifcrf.gov/, accessed on 11 January 2022) to conduct the (gene ontology) enrichment and Kyoto Encyclopedia of Genes and Genomes (KEGG) pathway analysis. We applied the results with *p* < 0.05 for the data analysis.

## Figures and Tables

**Figure 1 ijms-23-11590-f001:**
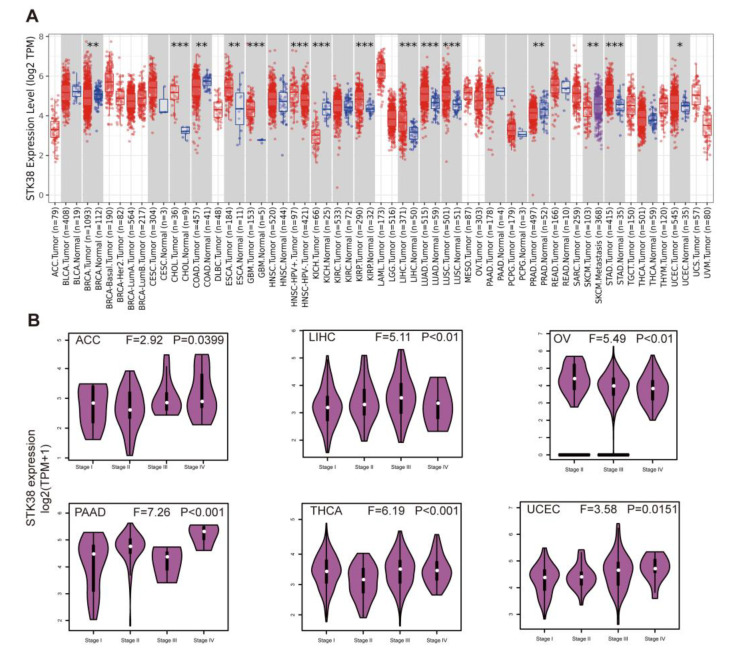
*STK38* expression profiles in normal tissues and cancers. (**A**) *STK38* expression level in TCGA tumors were analyzed using TIMER2.0 database (* *p* < 0.05; ** *p* < 0.01; *** *p* < 0.001). The red color represents tumor tissue samples, and the blue color represents normal samples. (**B**) Expression level of STK38 in main pathological stages (stage I, stage II, stage III, and stage IV) of ACC, LIHC, OV, PAAD, THCA and UCEC were analyzed using TCGA. Expression levels are shown as Log2 (TPM + 1).

**Figure 2 ijms-23-11590-f002:**
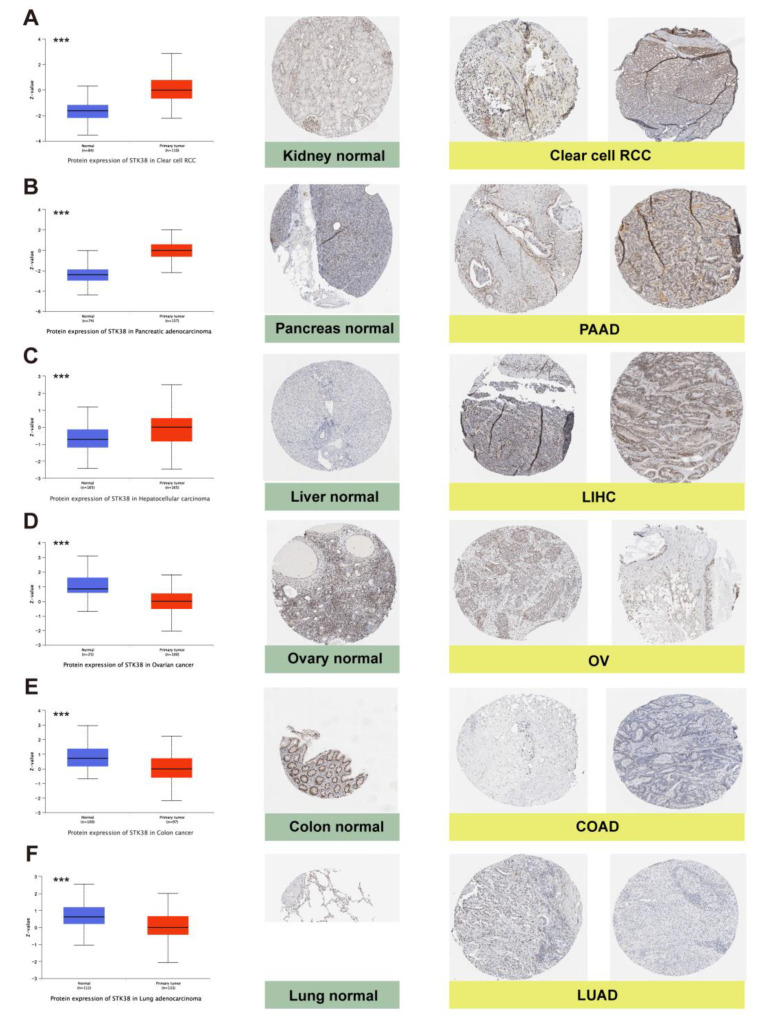
Protein expression levels of STK38 between normal tissues and tumor tissues (left), and representative immunohistochemical staining (IHC) in normal (middle) tissue and tumor (right) tissue samples in the HPA database. STK38 protein expression was significantly higher in clear cell RCC, PAAD and LIHC, and it was lower in OV, COAD and LUAD (**A**) Kidney. (**B**) Pancreas. (**C**) Liver. (**D**) Ovary. (**E**) Colon. (**F**) Lung. (*** *p* < 0.001). Protein expression data are originated from CPTAC database. Two immunohistochemistry images for each cancer were from different patients. Antibody CAB004673 was used for IHC. Scale bars: 200 μm.

**Figure 3 ijms-23-11590-f003:**
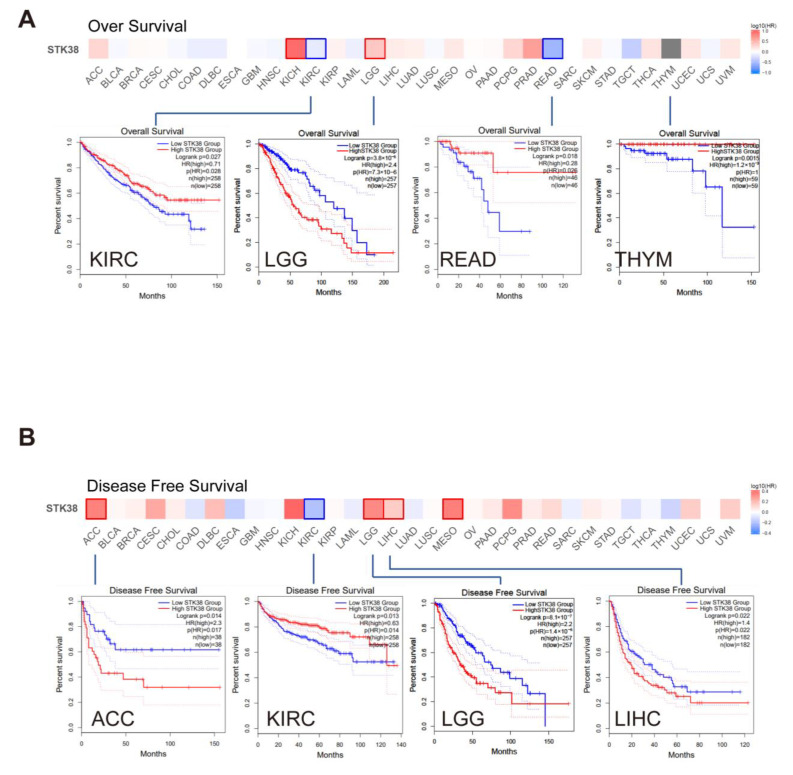
Correlation analysis between *STK38* gene expression and prognosis of different tumors in TCGA. GEPIA2 tool was used to perform overall survival (**A**) and disease-free survival (**B**) analyses of TCGA tumors by *STK38* gene expression. The survival map and Kaplan–Meier curves with positive results are listed. The red lines of Kaplan–Meier curves indicate the high *STK38* expression groups, and the blue lines indicate the low *STK38* expression groups.

**Figure 4 ijms-23-11590-f004:**
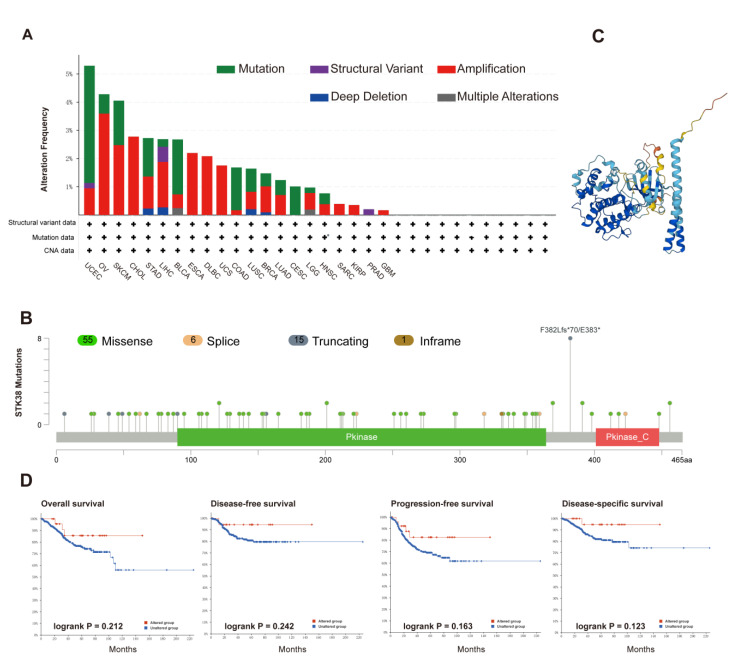
Alteration characterizes of *STK38* in TCGA tumors. The mutation features of *STK38* were analyzed using the cBioPortal tool. The alteration frequency with mutation type (**A**) and mutation site (**B**) are shown. (**C**) Protein structure of STK38. (**D**) Correlation analysis of *STK38* between mutation status and overall, disease-specific, disease-free and progression-free survival using the cBioPortal tool. The red lines represent the altered groups, and the blue lines represent the unaltered groups.

**Figure 5 ijms-23-11590-f005:**
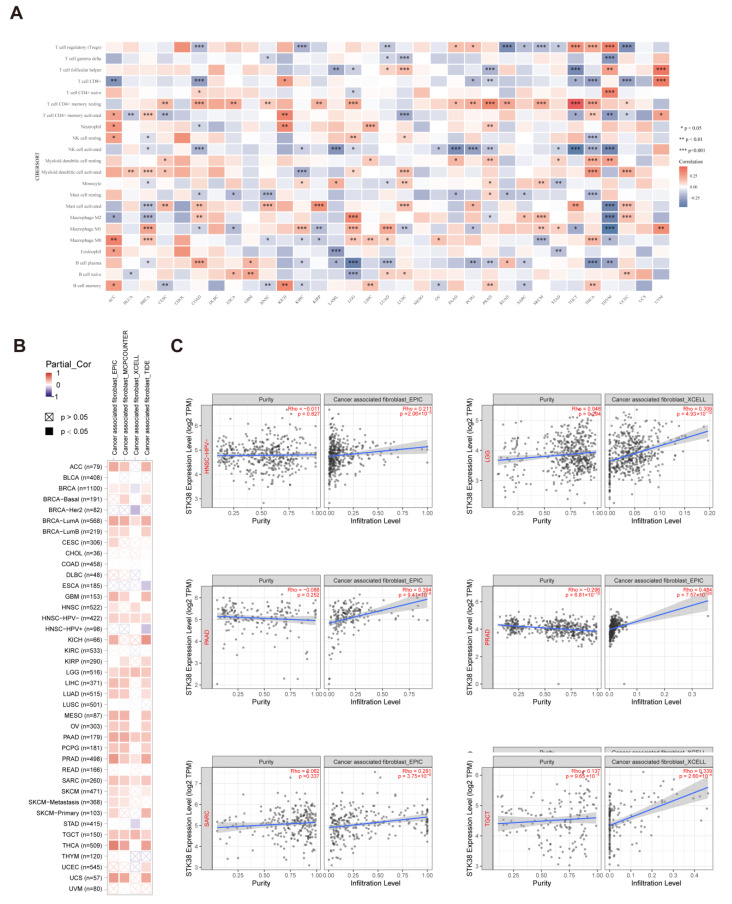
Relationship between *STK38* expression and immune infiltration (**A**) The expression of *STK38* is significantly correlated with a variety of immune cells infiltration levels based on CIBERSORT. * *p* < 0.05, ** *p* < 0.01, and *** *p* < 0.001. (**B**) The potential correlation analysis of *STK38* expression and immune infiltration of cancer-associated fibroblasts. The red color represents a positive relationship (0–1), while the blue color indicates a negative relationship (−1–0). The relationship with *p*-value < 0.05 is statistically significant. Statistically non-significant relationship are labeled with a cross. (**C**) The scatter plots of related cancers including HNSC-HPV-, LGG, TGCT, PRAD, SARC and PAAD.

**Figure 6 ijms-23-11590-f006:**
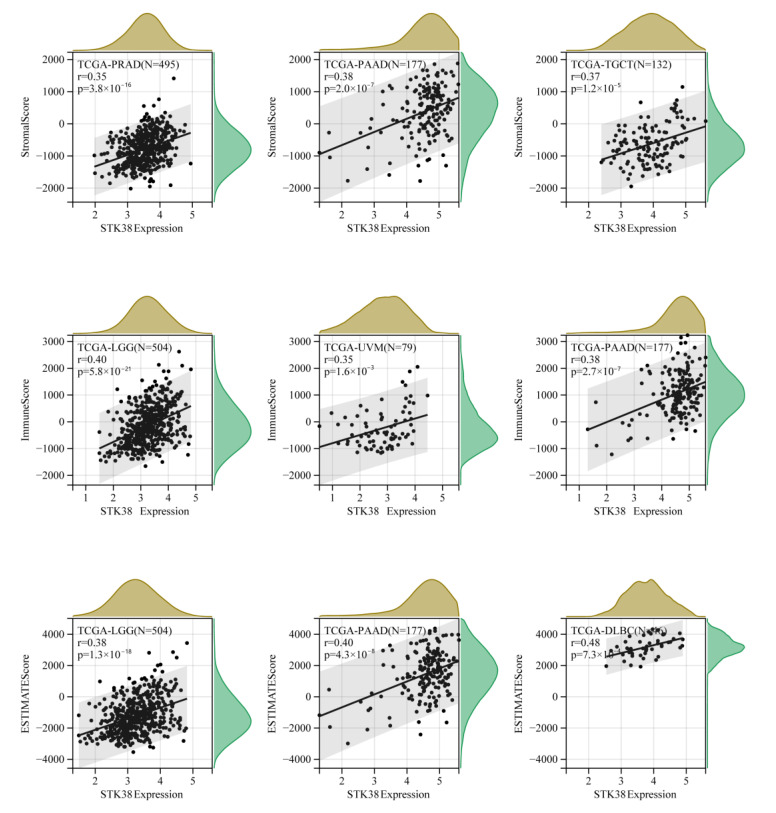
Top three scatter plots of correlation between *STK38* expression and ImmuneScore, StromalScore and ESTIMATEScore in multiple cancers.

**Figure 7 ijms-23-11590-f007:**
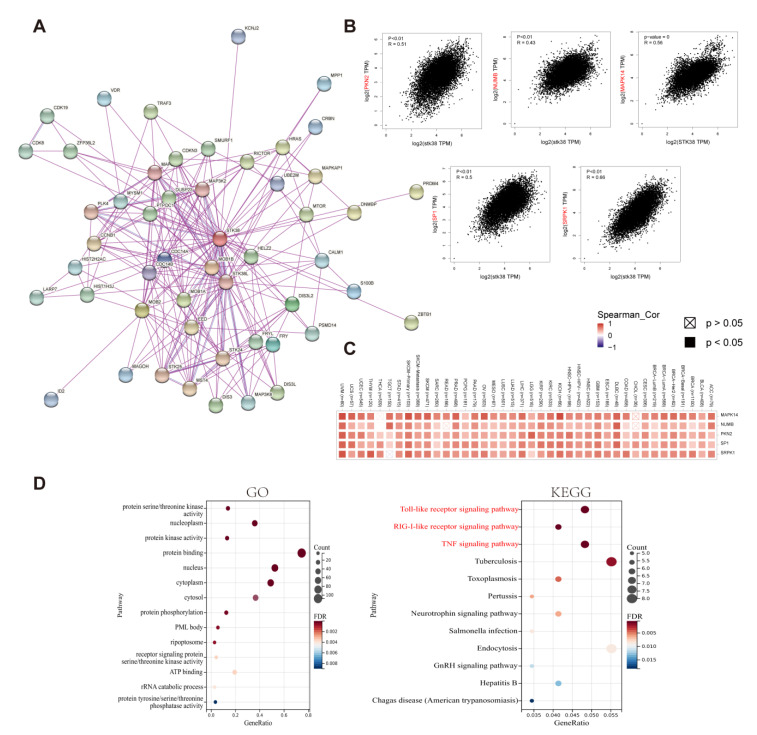
Enrichment and pathway analysis of STK38-related partners. (**A**) The experimentally verified STK38-binding proteins were obtained from the STRING website. (**B**) The correlation analysis between *STK38* and selected binding genes including PKN2, NUMB, MAPK14, SP1 and SRPK1 using the GEPIA2 tool. (**C**) The corresponding heatmap data of different cancer types are displayed in detail. (**D**) GO and KEGG pathway analysis were performed based on the *STK38*-binding and interacted genes.

**Table 1 ijms-23-11590-t001:** Association of *STK38* expression with DNA methylation.

Cancer	Methylation	
	*p*-Value	Correlation
ACC	0.082	0.197
BLCA	0.054	−0.095
**BRCA**	<0.001	−0.25
**CESC**	<0.001	−0.317
CHOL	0.438	−0.133
**COAD**	<0.001	−0.479
**READ**	<0.001	−0.378
DLBC	0.627	0.072
**ESCA**	0.02	−0.291
GBM	0.1	0.231
**HNSC**	<0.001	−0.176
KICH	0.401	−0.106
**KIRC**	0.011	−0.142
**KIRP**	0.019	0.142
**LAML**	0.047	0.188
**LGG**	0.008	−0.117
**LIHC**	<0.001	−0.272
**LUAD**	0.042	−0.096
**LUSC**	<0.001	−0.222
MESO	0.437	−0.085
OV	0.396	−0.393
**PAAD**	0.001	0.294
**PCPG**	<0.001	0.363
**PRAD**	<0.001	0.216
**SARC**	0.005	0.175
**SKCM**	<0.001	−0.218
STAD	0.033	−0.116
**TGCT**	<0.001	0.349
**THCA**	<0.001	−0.419
**THYM**	<0.001	0.548
UCEC	0.571	−0.027
**UCS**	0.001	−0.423
**UVM**	0.032	−0.241
**KIRC**	0.006	−0.156

The *p*-values in bold are statistically significant (less than 0.05).

## Data Availability

The data which support the findings of our study are openly available from the UALCAN (http://ualcan.path.uab.edu/analysisprot.html), Timer (http://timer.cistrome.org/) and STRING (https://string-db.org/) database.

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
