# Peer review of "Prognostic and Immunological Role of STK38 across Cancers: Friend or Foe?"

_ijms, 2022, doi:10.3390/ijms231911590_

Round 1
Reviewer 1 Report
Authors: Yankuo Liu, Zhiyuan Shi, Zeyuan Zheng et al.
Title: Prognostic and immunological role of STK38 across cancers: friend or foe?
COMMENTS:
This study was dedicated to the important and interesting subject in the field of molecular oncology. The Authors have obtained the intriguing data on a prognostic value of STK38 in different cancers and also revealed a role of STK38 in cancer immunity. There are a couple of points of minor criticism:
1. The Authors analyzed only the STK38 gene/protein expression. However, it would be nice if they say something about the functional activity of STK38. Was it possible to assess the level of STK38 kinase activity and how it correlated to the STK38 protein expression levels? Can STK38 kinase activity have a prognostic value as well?
2. It seems to me that the Authors' manner of citation in their manuscript does not correspondent to the rules of citation accepted in IJMS. It should be checked and corrected as numbers in brackets ([number]).
Reviewer 2 Report
Comments to authors:
The manuscript that was written by Dr. Yankuo Liu et al. is interesting to discuss possible roles of the STK38 in cancer cells, utilizing several data bases for analyses of its gene and protein expression. Authors have performed mainly in silico analyses of STK38 gene expression, its genetic alteration variants, and relationships with immune infiltration. However, it is obscure why authors have focused on the STK38 gene. If authors intended to publish the manuscript as a review article, it is essentially required why specifically the STK38 gene expression was selected to be examined. Before publication of the article, I strongly recommend authors to describe the background of the NDR/LATS family-encoding genes in detail, including the STK38. Moreover, a genetic evolution tree of the family genes or the protein structures of the family member could be illustrated. The localization of the STK38 protein would better be also commented. Such information would be a great help for readers comprehension.
Additionally, each figure legend needs essentially required information to be understood. If the figures were not explained enough, readers cannot understand them promptly.
Recommendation: Major revision
Minor comments
Page 3, L134-136, Figure 1: The reason why authors have selected ACC, LIHC, OV, PAAD, THCA, and DCED cells should be described. In the legend to Figure 1A, it is not described the meaning of colors (gray, red and blue). If they have no meaning, the figure should be drawn in white and black.
Page 3, L137: The data from UALCAN analysis is not shown.
Page 5, Figure 2: The description in the legend is not enough. Two photographs for each cancer cells are given. Are they originated from a same cancer of a patient? No scale bar is indicated. Are these photographs showing tissue staining or cellular staining? How STK38 was stained? In addition, the left part of the figure needs descriptions how they were analyzed.
Page 6, Figure 3: The description in the legend is not enough. Authors should give a note for colors. For example, what does it mean by red and blue colors? Is red indicating STK38 highly expressing group? Moreover, readers cannot understand the upper sides of the Figure 3A and 3B either.
Page 7, L202, Table 1: I could not find the name ESAD in Table 1. Both negative and positive correlations should be shown by under lines or bold typing. I recommend authors to add legend under the table, indicating notes to understand the results promptly.
Page 8: The reason why authors analyzed STK38 gene expression in immune cells is obscure. Does STK38 have some effects on immune functions?
Page 9, Figure 4: (A) The explanation for colors should be clearly shown. (B) Which mutation(s) can cause cell proliferation, differentiation, or cell death? Authors need to examine the known information for mutations on the STK38 gene. This figure seems to be indicating the amino acid changed sites on the STK38 protein by the mutations. Readers cannot understand the colors of indicators, yellow, blue, green, and brown. What do they mean by “Pkinase” and “Pkinase_C”? (C) Does it mean STK38-altered patients and non-altered by red and blue, respectively? Such information, which is not clearly shown in the figure, should be required to be noted in the legend.
Page 10, Figure 8: (B) Black blanks are not indicated.
Page 11, Figure 9: What it means by Stroma Score? What could be indicated by the relationships with SKT38 gene expression? Authors need to describe about that in the legend or the text.
I guess discussion will be completely edited with the information of the STK38 protein. Regarding the background, molecular (domain) structure, localization in a cell, expressing tissue or organs, should be re-examined. Moreover, which mutations on the protein can affect cancer should be noted. After the revision, I will evaluate the discussion section.
Other comments
Page 1, L31: Phenotypic and: type in plain text
Page 1, L42: breast: breast cancer?
Page 11, L255: STK38: type in italic. Authors should confirm that the gene names are typed in italic all through the text.
Reviewer 3 Report
The manuscript emphasizes the role of STK38 in several cancer types, including those with high mutational burden. The paper is well written, the authors used several database and biostatistics methods to sustain the validity of the results, based mainly on histopathology and molecular biology data. Authors provided a comprehensive image on STK38 significance in cancer prognosis and precision therapy, which will lead eventually to a better clinical outcome. The conclusions are sound; the way authors present the content is also flawless-all images are clear, very good resolution, including the supplemental images, they are no redundant data, the manuscript is well structured.
I strongly recommend accepting the paper.
Author Response
We are grateful to the reviewer for your effort reviewing our paper and your positive feedback.
Round 2
Reviewer 2 Report
Supplementary Figure S1 is missing. I will start to review the manuscript again when I received the totally revised one. Authors may also check all figures and text to confirm that it is completely freed from any errors including typos.
Author Response
Dear reviewer:
We appreciate the constructive comments, which have helped improve our manuscript. We have carefully considered the suggestions of reviewer and tried our best to make some changes in the manuscript. The red part that has been revised according to your comments. Revision notes, point-to-point, are given as follows:
Point 1: Supplementary Figure S1 is missing. I will start to review the manuscript again when I received the totally revised one. Authors may also check all figures and text to confirm that it is completely freed from any errors including typos.
Response 1: Thanks a lot for the reviewer's comments. We have added Figure S1 into the supplementary material. In addition, we have checked all figures and text carefully.

Round 3
Reviewer 2 Report
Comments to authors:
The manuscript that was written by Dr. Yankuo Liu et al has been successfully revised according to suggestions that I have commented before. Importantly, authors have edited explanations for essentially required Figures. That will be a good help for readers comprehension when the manuscript was published. This time, I would just give a suggestion for them to describe required descriptions especially about the software or homepage addresses of the internet access sites for in silico analyses. The description is required not only for the Methods section, but also for the legend to each Figure. Also, authors might better check all through the text again to confirm that it is completely freed from any errors, including typos.
